# Exon 11 Polymorphism (rs1126797) in the Thyroid Peroxidase (*TPO*) Gene Among Caucasian Polish Patients with Autoimmune Thyroiditis: A Short Communication

**DOI:** 10.3390/ijms26136299

**Published:** 2025-06-30

**Authors:** Katarzyna Lacka, Adam Maciejewski, Aleksandra M. Łącka, Waldemar Herman, Jan K. Lacki, Ryszard Żaba, Michał J. Kowalczyk

**Affiliations:** 1Department of Endocrinology, Metabolism and Internal Medicine, Poznan University of Medical Sciences, 60-355 Poznan, Poland; 2Department of Dermatology, Poznan University of Medical Sciences, 60-355 Poznan, Poland; 3Outpatients Unit for Endocrine Diseases, 67-400 Wschowa, Poland; 4Department of Medicine, The Jacob of Paradies University, 66-400 Gorzow Wielkopolski, Poland; 5Department of Internal Medicine, Collegium Medicum, University of Zielona Gora, 65-046 Zielona Gora, Poland

**Keywords:** *TPO* gene, polymorphism, autoimmune thyroiditis, rs1126797

## Abstract

Autoimmune thyroiditis (AIT), or Hashimoto’s thyroiditis, is one of the most prevalent autoimmune endocrine disorders. Its pathogenesis is complex and involves both environmental and genetic factors, yet it remains incompletely understood. Among the genetic contributors, thyroid-specific genes, including thyroid peroxidase (*TPO*) and thyroglobulin (*Tg*), have been implicated. The aim of this study was to investigate the potential association between the *TPO* gene single nucleotide polymorphism rs1126797, located in exon 11, and the risk of developing AIT in a Caucasian Polish population. To date, this SNP has not been studied in European cohorts, prompting us to explore its role following prior assessments in other ethnic groups. A total of 234 patients diagnosed with AIT and 132 healthy control subjects were enrolled. Genotyping of rs1126797 was performed using the TaqMan SNP Genotyping Assay. Allele and genotype frequencies were compared between groups, and associations with clinical parameters, including thyroid volume, were analyzed. No statistically significant differences in allele or genotype frequencies of rs1126797 were observed between AIT patients and healthy controls. However, a weak, significant trend was noted, suggesting a possible association between rs1126797 genotypes and thyroid volume in patients with AIT. Our findings do not support a significant role of the *TPO* rs1126797 polymorphism in conferring susceptibility to autoimmune thyroiditis in the studied Caucasian Polish population. However, the observed trend in thyroid volume among AIT patients with different rs1126797 genotypes warrants further investigation. Future studies involving larger and ethnically diverse cohorts are needed to validate these findings.

## 1. Introduction

Autoimmune thyroiditis (AIT), also known as Hashimoto’s thyroiditis, is one of the most prevalent autoimmune diseases overall and the most frequent among organ-specific autoimmune disorders [1]. Its global prevalence is estimated at approximately 5–7% of the general population, with a similar rate of around 5% reported in Poland [2].

Despite extensive research, the precise etiology of AIT remains incompletely understood. The condition results from a complex interplay of genetic, environmental, and epigenetic factors, classifying it as a multifactorial disease. Among the genetic contributors, both thyroid-specific and immune-related gene polymorphisms are implicated [3,4,5]. Although several genetic variants influencing immune regulation (e.g., HLA-DR3, PTPN22, CD40, FOXP3, CTLA-4, and cytokine genes) are clearly associated with autoimmunity, they are shared among different autoimmune diseases and do not entirely explain the thyroid-specific susceptibility [1,6].

The most extensively studied thyroid-specific genes in AIT include the *TPO* and *Tg* genes. The thyroid-stimulating hormone receptor (*TSHR*) gene has also been evaluated in AIT, although it is primarily associated with Graves’ disease and Graves’ orbitopathy [7]. Both thyroglobulin (Tg) and thyroid peroxidase (TPO) are major autoantigens in autoimmune thyroid diseases. Anti-TPO antibodies (TPOAb) are present in up to 90% of AIT patients, serving as a hallmark of the disease [8]. Beyond their diagnostic utility, these antibodies may also play a direct pathogenic role in disease development [9].

Several single nucleotide polymorphisms (SNPs) in the *TPO* gene have been investigated for their association with AIT pathogenesis. These include variants in the coding regions as well as the promoter region of the gene, though the findings across studies have often been inconsistent [10,11,12,13,14]. An intronic SNP, rs1126797, has previously been reported to be associated with thyroid autoimmunity and hypothyroidism [15,16,17]. Nevertheless, data on its genotype distribution in European populations are lacking.

This study aimed to evaluate the association between the exon 11 SNP of the *TPO* gene—rs1126797—and the development and clinical course of autoimmune thyroiditis in a Caucasian Polish population.

## 2. Results

There were no statistically significant differences in allele or genotype frequencies of the rs1126797 SNP between the AIT patients and the controls within the Caucasian Polish population. The CC genotype and C allele were slightly more frequent among the patients compared to the controls (43.59% vs. 40.91% and 66.24% vs. 64.02%, respectively), although these differences were not statistically significant (*p* > 0.05). The genotype distribution did not differ significantly under either the dominant (CC + CT vs. TT, *p* = 0.61) or recessive (CC vs. CT + TT, *p* = 0.62) inheritance models. Detailed genotype and allele frequencies are presented in Table 1.

In the subgroup of 116 AIT patients, further analysis was conducted to explore the association between rs1126797 genotypes and ultrasonographic features. No significant differences were observed in the genotype distribution between the patients with and without thyroid nodules (*p* > 0.05). However, thyroid volume varied significantly between the genotypes (*p* = 0.02). Median thyroid volumes were as follows: CC: 10.69 mL (range: 5.46–17.49), CT: 7.74 mL (range: 4.62–12.07), and TT: 13.11 mL (range: 7.31–17.71). The Dunn–Bonferroni post hoc test indicated that the most notable difference in thyroid volume was observed between the CT and TT genotypes, although it was still just a trend toward significance (*p* = 0.06).

## 3. Discussion

AIT has been the focus of numerous studies because of its multifactorial pathogenesis involving both non-modifiable (genetic) and modifiable (environmental) factors. Moreover, the association between AIT or anti-thyroid antibody positivity and other diseases is frequently analyzed. Beyond the typical symptoms of hypothyroidism observed in most patients, AIT has also been associated with an increased risk of mood disorders (predominantly bipolar disorder) [18], neuropsychiatric symptoms (also known as Hashimoto’s encephalopathy), and reduced quality of life despite biochemical euthyroidism [19,20]. Furthermore, TPOAb positivity has been linked to adverse pregnancy outcomes, including miscarriage, and anemia [21].

Understanding the underlying contributors to AIT is essential for identifying high-risk individuals, enabling earlier diagnosis, and ultimately elaborating preventive strategies. Multiple factors are already known, but the contribution of single factors is generally low [22,23]. Current consensus suggests that disease development arises from complex interactions between environmental triggers and a genetically determined immune predisposition. Among thyroid-specific genes, *TPO* and *TG* are natural candidates [9].

Thyroid peroxidase (TPO) is a 107 kDa glycosylated hemoprotein localized on the apical membrane of thyrocytes, where it plays a critical role in thyroid hormone synthesis [24]. The *TPO* gene, located on chromosome 2p25, comprises 17 exons [25]. Mutations in the *TPO* gene are the most common cause of dyshormonogenesis, which, in turn, is the second most common etiology of congenital hypothyroidism, often associated with goiter [26].

The current study did not find a significant association between the rs1126797 SNP and the risk of AIT. These findings are consistent with prior studies in Japanese and Egyptian populations [12,13]. Notably, Tomari et al. also failed to observe a correlation between this SNP and thyroid-specific antibody levels. Although rs1126797 is located in the coding region of the *TPO* gene (exon 11), it is a synonymous variant and does not result in an amino acid substitution [17]. Nevertheless, synonymous SNPs can still impact gene function by affecting mRNA stability, splicing efficiency, or translation dynamics. Moreover, the proximity of rs1126797 to other polymorphisms within the gene raises the possibility of linkage disequilibrium (LD) with functionally relevant variants. Table 2 summarizes the available data on rs1126797 in AIT patients (including our own results).

Although certain SNPs may not significantly influence disease risk, they can still modulate the clinical course, such as disease severity or age at diagnosis. We evaluated whether the rs1126797 SNP affects ultrasonographic features in patients with AIT. We observed that individuals with the CT genotype had a lower thyroid volume compared to those with the TT genotype, with borderline significance. CT vs. CC carriers also showed a non-significant trend toward smaller thyroid volume. This heterozygote-specific effect suggests a non-additive genetic model. These findings need further investigation in larger independent cohorts. Previous studies by Tomari et al. and Ahmed et al. also explored the association between rs1126797 and the clinical course of AIT but did not assess thyroid volume [12,13]. Other genetic variants, such as in *AATF*, *SMARCA2* or *PTPN22* genes, have previously been linked to thyroid volume in AIT [27,28].

Previous research has explored the role of rs1126797 in other thyroid-related conditions. Faam et al. showed that the C allele of this SNP (also referred to as T1936C) was associated with higher TPOAb levels in randomly selected participants [15]. Ghanooni et al. identified rs1126797 as part of a haplotype block significantly associated with TPOAb positivity, though the specific contribution of this SNP remained uncertain [16]. Balmiki et al. reported that the T allele of rs1126797 was protective against hypothyroidism in adults [17]. However, Su et al. found no significant association between rs1126797 and congenital hypothyroidism due to dyshormonogenesis in a Chinese population [29].

Several other *TPO* gene SNPs have been studied in relation to AIT. Promoter region (e.g., rs2071399, rs2071400, rs2071402, rs2071403) and exon 12 (rs732608 and rs732609) SNPs have been investigated in diverse populations [12,13,14,30]. Other regions of the *TPO* gene have also been analyzed, further supporting the gene’s role in thyroid autoimmunity [10,12]. In Egyptian patients, rs2071400 C/T and rs732609 A/C were shown to be associated with an increased risk of AIT [13]. In an Iranian population, rs732609 and another exon 12 SNP, rs732608, were found to be associated with subclinical hypothyroidism [30]. Japanese studies found distinct genotype distributions for rs2071400 and rs2071403 between AIT cases and controls, and reported associations of rs2071400 and rs2048722 with TPOAb levels [12]. The rs2071403 SNP was identified as being associated with TPOAb positivity in a Korean genome-wide association study (GWAS), although no link to hypothyroidism was found [31]. In Iranian populations, Faam et al. and Khoshi et al. demonstrated associations between other *TPO* polymorphisms (Asn698Thr/T1936C exon 11, Thr725Pro/A2257C exon12) and serum TPOAb levels [15,30]. Ghanooni et al. highlighted rs6605278 in the 3’UTR as the strongest contributor to TPOAb positivity among multiple *TPO* SNPs [16].

Studies on Caucasian AIT populations are more limited. Brcic et al. found that rs11675434 was significantly associated with both AIT risk and TPOAb levels [10], a finding supported by studies in Chinese and other Western populations (Sweden, USA, and Australia) [32,33,34,35]. Our previous investigation of promoter SNPs (rs2071399, rs2071400, rs2071402, rs2071403) in Polish AIT patients did not show significant associations [14]. Jabrocka-Hybel et al. reported a weak association for rs11211645 (upstream region) but found no relevance for other variants (rs961028 and rs2276704) [11].

Given the overlapping autoimmune pathophysiology between AIT and Graves’ disease (GD) or Graves’ orbitopathy (GO), some *TPO* SNPs have also been explored in these conditions. While rs11675434 was not associated with GD in a Polish cohort, it was linked to the presence and severity of GO [36,37]. In Japanese patients, rs2071400 and rs2071403 were associated with GD susceptibility [12].

The rs1126797 SNP was selected for analysis because of its limited prior investigation in Caucasian populations. Previous studies in Asian populations suggested an association between this SNP and TPOAblevels or subclinical hypothyroidism, warranting its evaluation in a different ethnic context [15,16,17]. Moreover, SNPs with a relatively balanced allele distribution are better candidates for genetic predisposition studies. This is a truth for rs1126797 among Caucasians (C allele = 0.64 vs. T allele = 0.36).

This study has several limitations that should be acknowledged. First, the relatively modest sample size, particularly of the control group, may have limited the statistical power to detect weak associations. Second, although the study population was ethnically homogeneous, it was restricted to Caucasian individuals from a single geographic region in Poland, which may limit the generalizability of the findings to other populations. Third, the functional impact of the rs1126797 variant on *TPO* gene expression or protein function is uncertain. Finally, environmental and epigenetic factors, which are known contributors to autoimmune thyroiditis, were not fully evaluated and may have influenced disease susceptibility or phenotype.

## 4. Materials and Methods

### 4.1. Patients

The study group included 234 unrelated patients with AIT of Caucasian Polish origin (224 women and 10 men), with a mean age of 46.88 ± 14.19 years (range: 18–75). The diagnosis of AIT was based on elevated levels of TPOAb and/or anti-Tg antibodies, a characteristic ultrasound image of the thyroid gland, and either a hypo- or euthyroid state determined by TSH levels (standard diagnostic criteria, as described previously) [14]. At the time of this study, all patients were euthyroid under L-thyroxine substitution therapy.

The control group consisted of 132 healthy individuals (104 women and 28 men) with no personal or family history of thyroid dysfunction, goiter, autoimmune thyroid disease (AITD), or other autoimmune disorders. Their mean TSH level was 1.57 ± 1.02 mIU/L, and the mean age was 33.96 ± 14.22 years (range: 18–74). The controls were recruited from the same geographical region as the AIT patients.

Detailed and standardized ultrasound reports from the same medical center, all performed by the same ultrasonographer, were available for 116 patients.

This study was approved by the local ethics committee of Poznan University of Medical Sciences (approval no. 443/17), and written informed consent was obtained from all the participants, both the patients and the controls.

### 4.2. Methods

#### Genotyping of *TPO* Exon 11 SNP (rs1126797)

Genomic DNA was extracted from peripheral blood mononuclear cells using either the NucleoSpin Blood Kit (Macherey-Nagel, Düren, Germany) or the QIAamp DNA Blood Mini Kit (QIAGEN, Hilden, Germany), according to the manufacturers’ protocols.

Genotyping was performed using the TaqMan SNP Genotyping Assay (cat. no. 4351379, assay ID: C___8760351_10, Thermo Fisher Scientific, Waltham, MA, USA) and Fast Probe qPCR Master Mix 2× (cat. no. E0422, EURx, Gdańsk, Poland). Reactions were carried out using the LightCycler 2.0 capillary thermal cycler (Roche Diagnostics, Mannheim, Germany) following the protocol detailed in Table 3. Genotypes were determined by analyzing the ratio of fluorescence signals at 530 nm and 560 nm, as illustrated in Figure 1. All batches were cross-calibrated, and 18% of the samples were randomly duplicated to confirm genotyping consistency.

### 4.3. Statistical Analysis

The Hardy–Weinberg equilibrium was assessed using the chi-squared test (χ^2^) for both groups. Differences in allele and genotype frequencies between the AIT patients and the controls were analyzed using the χ^2^ or χ^2^ test for trend, as appropriate. A *p*-value < 0.05 was considered statistically significant. Odds ratios (ORs) with 95% confidence intervals (CIs) were calculated to estimate the strength of associations.

To assess differences in thyroid volume between genotype groups, the Kruskal–Wallis test was used, followed by the Dunn–Bonferroni post hoc test. Statistical analyses were performed using PQStat version 1.8.6 (PQStat Software, Poznań, Poland).

## 5. Conclusions

This study represents the first investigation of the rs1126797 SNP located in exon 11 of the *TPO* gene in a Caucasian Polish population with autoimmune thyroiditis. Although no significant association was found between this SNP and the overall risk of AIT, a suggestive trend was observed between rs1126797 genotypes and thyroid volume in affected individuals. These findings indicate that while rs1126797 may not be a major risk factor for AIT susceptibility, it could have a modulatory role in disease phenotype. Given the multifactorial nature of AIT and the genetic heterogeneity observed across different ethnic groups, further studies in larger and more diverse populations are warranted.

## Figures and Tables

**Figure 1 ijms-26-06299-f001:**
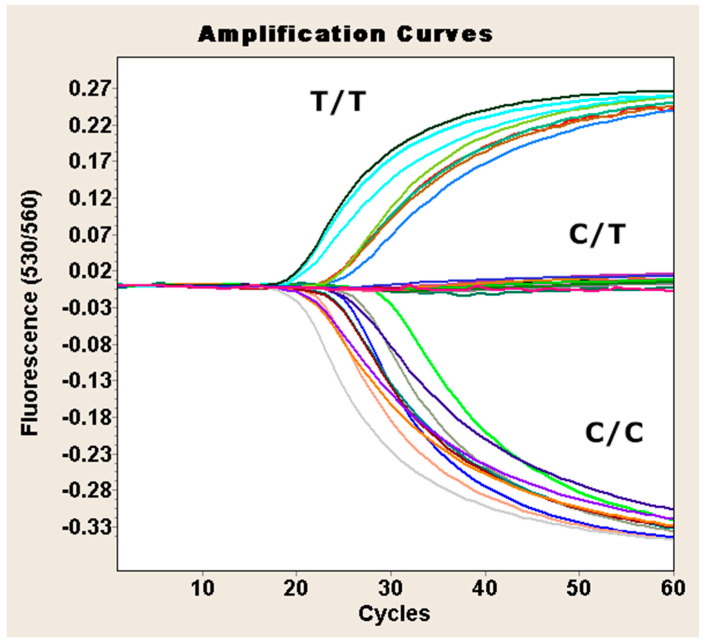
Genotyping of the rs1126797 SNP of the *TPO* gene using TaqMan real-time PCR. The *X*-axis represents PCR cycle number; the *Y*-axis shows the fluorescence ratio. TT, CT, and CC represent the rs1126797 genotypes.

**Table 1 ijms-26-06299-t001:** Allele and genotype frequencies of the exon 11 *TPO* single nucleotide polymorphism (rs1126797) in the patients with autoimmune thyroiditis (AIT) and the healthy controls in the Polish population.

SNP	Allele/Genotype	AIT (n = 234), n (%)	Controls (n = 132), n (%)	*p*-Value	OR (95% CI)
rs1126797	Genotypes				
	CC	102 (43.59)	54 (40.91)	0.83 * 0.54 **	
	CT	106 (45.30)	61 (46.21)		
	TT	26 (11.11)	17 (12.88)		
	CC vs. CT + TT	102 (43.59) vs. 132 (56.41)	54 (40.91) vs. 78 (59.09)	0.62 *	1.12 (0.72–1.72)
	CC + CT vs. TT	208 (88.89) vs. 26 (11.11)	115 (87.12) vs. 17 (12.88)	0.61 *	1.18 (0.62–2.27)
	Alleles				
	C allele	310 (66.24)	169 (64.02)	0.54 *	1.10 (0.80–1.51)
	T allele	158 (33.76)	95 (35.98)		

Values in parentheses indicate percentages of the group. * Chi-squared test; ** chi-squared test for trend. SNP—single nucleotide polymorphism. OR—odds ratio. CC, CT, and TT represent the rs1126797 genotypes.

**Table 2 ijms-26-06299-t002:** A literature review on exon 11 *TPO* gene single nucleotide polymorphisms (rs1126797) in autoimmune thyroiditis (AIT).

Author	Population	Statistics
	Japanese 147 AIT and 92 healthy controls	- NS. - No association with TPOAb level.
Ahmed et al., 2021 [13]	Egyptian 200 AIT and 100 healthy controls	- NS. - No association with the disease severity.
Lacka et al. [present study]	Polish 234 AIT and 132 healthy controls	- NS. - Weak association with thyroid volume.

NS—not significant = no association with the disease risk.

**Table 3 ijms-26-06299-t003:** Protocol for preparation of a PCR reaction mix and PCR conditions for *TPO* gene exon 11 SNP (rs1126797) genotyping using the TaqMan assay.

**PCR Reaction Mix (Total Volume: 20 μL)**		
Component		Volume (μL)
Fast Probe qPCR Master Mix 2×		10.0
Nuclease-Free Water		8.5
TaqMan SNP Genotyping Assay		0.5
Genomic DNA		1.0
**PCR Cycling Conditions**		
Step	Temperature (°C)	Time
Predenaturation	95	10 min
Denaturation	92	15 s (at least 40 cycles)
Annealing/Extension	60	60 s
Final Extension	60	10 min

## Data Availability

The data are included in the article. Additional information is available from the corresponding author upon request.

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
