# Peer review of "Exon 11 Polymorphism (rs1126797) in the Thyroid Peroxidase (TPO) Gene Among Caucasian Polish Patients with Autoimmune Thyroiditis: A Short Communication"

_ijms, 2025, doi:10.3390/ijms26136299_

Round 1
Reviewer 1 Report
Comments and Suggestions for Authors
The article examines the correlation of polymorphism rs1126797 in exon 11 of the thyroid peroxidase (TPO) gene with the occurrence of autoimmune thyroiditis (AIT) in the Polish Caucasian population. A total of 234 patients with AIT and 132 healthy individuals who comprised the control group were involved in this work. Although no statistically significant difference in allele or genotype frequencies between the two groups could be found, a trend towards association of the polymorphism with thyroid gland volume was observed. Thus, this SNP is not revealed to be a significant risk factor for AIT, but it might influence the clinical presentation of the disease. Special attention is drawn to additional studies on larger and more heterogeneous populations.
In my opinion, this article provides a valuable addition to the understanding of genetic factors that may be associated with AIT, especially in Caucasian populations, where the frequency of the polymorphisms examined varies from that of other ethnic groups. It is also the first time rs1126797 has been examined in this setting. Even though the outcome was negative regarding association with AIT risk, the trend identified with tumor of the thyroid gland can pave the way for novel lines of investigation. This indicates a potential modifying effect of the polymorphism upon the disease phenotype, and this could have implications at the clinical level regarding prognosis or treatment strategy.
The essay requires significant restructuring and inclusion of additional experimental research.
- As the rs1126797 polymorphism is a synonymous one, it does not alter the amino acid sequence but may still have an influence on mRNA stability or translation efficiency. A good experimental approach would be to express the plasmids containing the various gene variants (TT, CT, CC) in thyroid cells and compare their influence on TPO mRNA levels (measured by qPCR) and the respective protein (analyzed by Western blot or ELISA).
- Because of the potential for linkage disequilibrium with nearby functional SNPs, more extensive genotyping of the region surrounding exon 11 is recommended, to assess whether rs1126797 is part of particular haplotypes that are linked to AIT. This can be done by using NGS or SNP arrays.
- Based on the association that has been observed between the polymorphism and thyroid tumors, it would be pertinent to investigate if there is any correlation of rs1126797 with inflammation markers or lymphocyte differentiation in the thyroid gland. It is recommended that tissue samples be obtained from AIT patients of varying genotypes and immunohistochemistry staining done for CD3, CD20, and IL-6 markers.
- To validate the trend established, we recommend that the study be repeated in other European populations (e.g., southern European) or non-European groups (e.g., African or Asian) with larger cohorts. Preferably, having a meta-analysis that combines data would enhance the level of evidence.
- The abstract provides a good description of the aims and results; however, it could be enhanced by greater concision. Mention of the relevance of TPO and Tg could be curtailed to allow more room for an exploration of the rationale for the selection of rs1126797 and the implication of the results. This would render the abstract more effective as a brief guide for the reader, as opposed to a lengthy introduction.
- The introduction satisfactorily addresses the etiological dimensions of the disease as well as the multifactorial nature of genetic factors, yet the linkage of this polymorphism to existing knowledge appears to be somewhat lagging. A preliminary mention of rs1126797 and the reasons why it is being particularly studied in the Caucasian population would render the argument more logical. In addition, a more specific thematic emphasis would strengthen the introduction.
- The methods section is well described; however, specific details of the PCR protocol, i.e., temperatures and duration, might be put in an Appendix or Table. This would help make the main text easier to read for non-specialist readers, while still being friendly to replication by specialists.
- The results are presented in a clear manner and are founded upon suitable statistical methods. Table 1, however, which gives genotype and allele frequencies, would benefit from additional textual explanation to facilitate understanding. A short interpretation of trends, e.g., p-value of 0.06, within the main text would give the reader a better guide.
- The argument is clearly well-researched and well-referenced, if somewhat long and, in certain areas, repeating what had been previously stated. The comparative point to other research is a good one, but it would be advantageous to bring the findings together in a summary table or brief interpretive model. That would enable the reader to see at a glance the contribution of this research to the existing literature.
Author Response
Reviewer 1
The article examines the correlation of polymorphism rs1126797 in exon 11 of the thyroid peroxidase (TPO) gene with the occurrence of autoimmune thyroiditis (AIT) in the Polish Caucasian population. A total of 234 patients with AIT and 132 healthy individuals who comprised the control group were involved in this work. Although no statistically significant difference in allele or genotype frequencies between the two groups could be found, a trend towards association of the polymorphism with thyroid gland volume was observed. Thus, this SNP is not revealed to be a significant risk factor for AIT, but it might influence the clinical presentation of the disease. Special attention is drawn to additional studies on larger and more heterogeneous populations.
In my opinion, this article provides a valuable addition to the understanding of genetic factors that may be associated with AIT, especially in Caucasian populations, where the frequency of the polymorphisms examined varies from that of other ethnic groups. It is also the first time rs1126797 has been examined in this setting. Even though the outcome was negative regarding association with AIT risk, the trend identified with tumor of the thyroid gland can pave the way for novel lines of investigation. This indicates a potential modifying effect of the polymorphism upon the disease phenotype, and this could have implications at the clinical level regarding prognosis or treatment strategy.
The essay requires significant restructuring and inclusion of additional experimental research.
1. As the rs1126797 polymorphism is a synonymous one, it does not alter the amino acid sequence but may still have an influence on mRNA stability or translation efficiency. A good experimental approach would be to express the plasmids containing the various gene variants (TT, CT, CC) in thyroid cells and compare their influence on TPO mRNA levels (measured by qPCR) and the respective protein (analyzed by Western blot or ELISA).
Response: Thank you for your valuable remark. We fully agree that this aspect warrants further exploration to verify the potential functional impact of the SNP. However, as our study is primarily clinically oriented, we do not currently have access to the experimental facilities required for such investigations. We hope that future studies will help clarify the biological relevance of this polymorphism
2. Because of the potential for linkage disequilibrium with nearby functional SNPs, more extensive genotyping of the region surrounding exon 11 is recommended, to assess whether rs1126797 is part of particular haplotypes that are linked to AIT. This can be done by using NGS or SNP arrays.
Response: We appreciate this insightful suggestion. We agree that other SNPs in linkage disequilibrium with rs1126797 could potentially contribute to the observed associations. However, a comprehensive assessment of LD patterns would indeed require high-throughput genotyping methods, such as direct sequencing, SNP arrays, or NGS, which were beyond the scope of our current study.
It is worth noting that Balamiki et al. 2014 previously analyzed LD between rs1126797 and several neighboring SNPs located in exons 10, 12, and intron 11 of the TPO gene, and found no evidence of linkage disequilibrium. Nevertheless, we agree that further detailed haplotype analysis using more extensive genotyping approaches may provide additional insights.
3. Based on the association that has been observed between the polymorphism and thyroid tumors, it would be pertinent to investigate if there is any correlation of rs1126797 with inflammation markers or lymphocyte differentiation in the thyroid gland. It is recommended that tissue samples be obtained from AIT patients of varying genotypes and immunohistochemistry staining done for CD3, CD20, and IL-6 markers.
Response: We fully agree that analyzing tissue-level immunological parameters could provide valuable insights into the potential functional implications of this SNP. However, due to the retrospective nature of our study and the lack of access to thyroid tissue samples from the included patients, it is not feasible to perform such analyses within the current project. Nonetheless, we recognize the value of this approach and believe it represents an excellent direction for a future, prospectively designed study focused on the interaction between TPO gene polymorphisms (together with different gene SNPs) and immune or inflammatory responses in autoimmune thyroid disease.
4. To validate the trend established, we recommend that the study be repeated in other European populations (e.g., southern European) or non-European groups (e.g., African or Asian) with larger cohorts. Preferably, having a meta-analysis that combines data would enhance the level of evidence.
Response: We appreciate your valuable suggestion regarding the need for validation in diverse populations. We fully agree that replication of our findings in larger cohorts, particularly from other European regions (e.g., Southern Europe) or non-European populations (e.g., African or Asian), would be essential to strengthen the evidence for any association involving rs1126797.
As noted in the Discussion, to our knowledge, no studies have investigated this SNP specifically in European populations. While there are a few reports exploring rs1126797 in the context of autoimmune thyroiditis in other populations, these studies did not assess its potential impact on thyroid volume or other ultrasonographic features. We also agree that future meta-analyses combining data across multiple cohorts would be highly beneficial in evaluating the broader relevance of this polymorphism.
5. The abstract provides a good description of the aims and results; however, it could be enhanced by greater concision. Mention of the relevance of TPO and Tg could be curtailed to allow more room for an exploration of the rationale for the selection of rs1126797 and the implication of the results. This would render the abstract more effective as a brief guide for the reader, as opposed to a lengthy introduction.
Response: Thank you for this suggestion. We have revised the abstract to reduce background information and more clearly highlight the motivation behind SNP selection (changes marked red).
6. The introduction satisfactorily addresses the etiological dimensions of the disease as well as the multifactorial nature of genetic factors, yet the linkage of this polymorphism to existing knowledge appears to be somewhat lagging. A preliminary mention of rs1126797 and the reasons why it is being particularly studied in the Caucasian population would render the argument more logical. In addition, a more specific thematic emphasis would strengthen the introduction.
Response: Thank you for the suggestion. We have revised the Introduction to include a more explicit rationale for focusing on the rs1126797 polymorphism. The changes are marked red for clarity.
7. The methods section is well described; however, specific details of the PCR protocol, i.e., temperatures and duration, might be put in an Appendix or Table. This would help make the main text easier to read for non-specialist readers, while still being friendly to replication by specialists.
Response: Thank you for the suggestion. We have moved the detailed PCR protocol, including temperatures and cycle durations, to a separate table (Table 1). This should improve clarity.
8. The results are presented in a clear manner and are founded upon suitable statistical methods. Table 1, however, which gives genotype and allele frequencies, would benefit from additional textual explanation to facilitate understanding. A short interpretation of trends, e.g., p-value of 0.06, within the main text would give the reader a better guide.
Response: We have improved and extended the explanation related to Table 1 in the Results section. We believe this enhances the clarity and interpretability of the findings for the reader. All changes have been marked red.
9. The argument is clearly well-researched and well-referenced, if somewhat long and, in certain areas, repeating what had been previously stated. The comparative point to other research is a good one, but it would be advantageous to bring the findings together in a summary table or brief interpretive model. That would enable the reader to see at a glance the contribution of this research to the existing literature.
Response: We thank the Reviewer for this remark. In response, we have reorganized the Discussion section to improve clarity and eliminate repetitive elements. Additionally, we would like to note that a summary table (Table 3) comparing key findings on the rs1126797 SNP in AIT patients from previous studies had already been included in the manuscript, we have added our own results.
Reviewer 2 Report
Comments and Suggestions for Authors
This manuscript addresses whether the rs1126797 SNP in exon 11 of the thyroid peroxidase gene is associated with autoimmune thyroiditis (AIT) in a Caucasian-Polish population. The study is well-executed, with a clearly defined clinical cohort, appropriate genotyping methods, and a reasonable analytic approach. The authors find that rs1126797 is not associated with risk for AIT. This is a meaningful negative result, especially given prior inconsistent reports across populations. The authors place this result appropriately in the context of the literature, and the manuscript is well-written and clear.
My biggest issue, however, is that the manuscript underdevelops an exploratory finding that emerged from their analysis: a weak but statistically suggestive association between rs1126797 genotype and thyroid volume. This analysis is relegated to a brief paragraph in the Results and is not discussed in the Discussion section at all. This is a missed opportunity, particularly because it is the only marginally significant result in the study. Moreover, the authors report an unusual pattern in these findings, the CT heterozygotes had the smallest volumes and TT homozygotes the largest. This pattern deviates from a linear gene-dosage model and is not immediately biologically intuitive. This warrants more careful interpretation, if not a fuller reevaluation.
I recommend either expanding the discussion of the thyroid volume result to acknowledge its implications and limitations. The paper's main conclusion (that rs1126797 does not contribute significantly to AIT risk) is an important result and contributes incrementally to the field. The manuscript would be strengthened by a more cohesive framing of its secondary findings, however.
Author Response
Reviewer 2.
This manuscript addresses whether the rs1126797 SNP in exon 11 of the thyroid peroxidase gene is associated with autoimmune thyroiditis (AIT) in a Caucasian-Polish population. The study is well-executed, with a clearly defined clinical cohort, appropriate genotyping methods, and a reasonable analytic approach. The authors find that rs1126797 is not associated with risk for AIT. This is a meaningful negative result, especially given prior inconsistent reports across populations. The authors place this result appropriately in the context of the literature, and the manuscript is well-written and clear.
My biggest issue, however, is that the manuscript underdevelops an exploratory finding that emerged from their analysis: a weak but statistically suggestive association between rs1126797 genotype and thyroid volume. This analysis is relegated to a brief paragraph in the Results and is not discussed in the Discussion section at all. This is a missed opportunity, particularly because it is the only marginally significant result in the study. Moreover, the authors report an unusual pattern in these findings, the CT heterozygotes had the smallest volumes and TT homozygotes the largest. This pattern deviates from a linear gene-dosage model and is not immediately biologically intuitive. This warrants more careful interpretation, if not a fuller reevaluation.
I recommend either expanding the discussion of the thyroid volume result to acknowledge its implications and limitations. The paper's main conclusion (that rs1126797 does not contribute significantly to AIT risk) is an important result and contributes incrementally to the field. The manuscript would be strengthened by a more cohesive framing of its secondary findings, however.
Response:
We sincerely appreciate your positive evaluation of our manuscript.
Thank you in particular for highlighting the exploratory finding related to thyroid volume and rs1126797 SNP genotype. We agree that, although secondary, this observation may hold clinical relevance. We have expanded the Discussion section to address this point more thoroughly (paragraph marked red).
Our initial caution in concluding stemmed from the relatively small sample size and the absence of a direct explanation for this association. Therefore, we believe that further studies are necessary to confirm this finding before more definitive conclusions can be made.
We hope these additions enhance the manuscript by offering a more comprehensive interpretation of our findings, while maintaining a clear focus on the primary conclusion—that rs1126797 is not significantly associated with AIT risk in the studied population.
Round 2
Reviewer 1 Report
Comments and Suggestions for Authors
The authors have adequately revised their manuscript. I recommend publication of the manuscript.